# Structural Characterization and Antidepressant-like Effects of *Polygonum sibiricum* Polysaccharides on Regulating Microglial Polarization in Chronic Unpredictable Mild Stress-Induced Zebrafish

**DOI:** 10.3390/ijms25042005

**Published:** 2024-02-07

**Authors:** Yingyu Zhang, Danyang Wang, Jiameng Liu, Yajuan Bai, Bei Fan, Cong Lu, Fengzhong Wang

**Affiliations:** 1Institute of Food Science and Technology, Chinese Academy of Agricultural Sciences (CAAS), Beijing 100193, China; zhangyy0427@163.com (Y.Z.); wdanyang1111@163.com (D.W.);; 2Sanya Institute, Hainan Academy of Agricultural Sciences, Sanya 572025, China

**Keywords:** depression, *Polygonati Rhizoma* polysaccharides, microglia polarization

## Abstract

Polysaccharides are one of the main active ingredients of *Polygonum sibiricum* (PS), which is a food and medicine homolog used throughout Chinese history. The antidepressant-like effects of PSP and its underlying mechanisms remain elusive, especially the regulation of microglial polarization. The current study determined the chemical composition and structural characteristics of PSP. Then, the chronic unpredictable mild stress (CUMS) procedure was carried out on the zebrafish for 5 weeks, and PSP was immersed for 9 days (1 h/d). The body weight of zebrafish was monitored, and behavioral tests, including the novel tank test and light and dark tank test, were performed to evaluate the antidepressant-like effects of PSP. Then, the function of the hypothalamic-pituitary-interrenal (HPI) axis, the levels of peripheral inflammation, neuronal and blood–brain barrier damage in the mesencephalon and telencephalon, and the mRNA expression of M1/M2 phenotype genes in the brain were examined. PSP samples had the typical structural characteristics of polysaccharides, consisting of glucose, mannose, and galactose, with an average Mw of 20.48 kDa, which presented porous and agglomerated morphologies. Compared with untreated zebrafish, the depression-like behaviors of CUMS-induced zebrafish were significantly attenuated. PSP significantly decreased the levels of cortisol and pro-inflammatory cytokines and increased the levels of the anti-inflammatory cytokines in the body of CUMS-induced depressive zebrafish. Furthermore, PSP remarkably reversed the neuronal and blood–brain barrier damage in the mesencephalon and telencephalon and the mRNA expression of M1/M2 phenotype genes in the brain. These findings indicated that the antidepressant-like effects of PSP were related to altering the HPI axis hyperactivation, suppressing peripheral inflammation, inhibiting neuroinflammation induced by microglia hyperactivation, and modulating microglial M1/M2 polarization. The current study provides the foundations for future examinations of PSP in the functional foods of emotional regulation.

## 1. Introduction

Depression has become one of the most common types of psychological disorders, characterized by prolonged depressed mood, especially in the outbreak of the new COVID-19 and post-epidemic era environment, which severely limits people’s psychosocial abilities and reduces the quality of life [1]. Chemically synthesized drugs are the main clinical drugs currently used for depression, which not only take longer to produce efficacy but also are accompanied by severe and serious side effects such as sudden weight loss, anorexia, and insomnia, making their safety a potential problem and limiting their clinical application [2]. With the upgrading of the consumption structure and demand for nutrition and health, functional foods of emotional health are in robust demand [3]. It is especially important to develop health products with high antidepressant effects to prevent and treat depression.

Neuroinflammation is considered a pathological mechanism of depression [4]. Microglia are key members of the immune defense in the central nervous system (CNS), which plays a central role in immunity and inflammation [5]. When homeostasis is disturbed, microglia can be rapidly activated. The activated microglia have a “double-edged” effect, manifesting themselves in two polarized phenotypes: a classically activated M1 and an alternatively activated M2 phenotype. M1 microglia are associated with the excessive release of pro-inflammatory factors that can exacerbate inflammatory damage in the CNS and lead to neurotrophic dysfunction, while M2 microglia are associated with the secretion of anti-inflammatory and neurotrophic factors that can help to antagonize inflammatory damage and promote recovery and remodeling of neural tissue [6,7,8]. One of the potential therapeutic targets for depression is regulating the M1/M2 polarization of microglia to dynamically regulate neuroinflammation and maintain homeostasis [9].

*Polygonum sibiricum* (PS) is a traditional food and medicine homolog in China. *Polygonum sibiricum* polysaccharides (PSP) are one of the main active components, known for their low toxicity and suitability for long-term administration, making them potentially useful in the healthcare and food industries [10]. In previous studies, PSP showed antidepressant effects in both lipopolysaccharide-induced and chronic unpredictable mild stress (CUMS)-induced depressive mice models. The regulation of the hypothalamic–pituitary–adrenal (HPA) axis dysfunction, a neurotransmitter (5-hydroxytryptamine, 5-HT and norepinephrine, NE) levels, neuroinflammation, gut microbiota composition, and short-chain fatty acids levels were the antidepressant mechanisms involved [11,12,13,14]. However, the mechanisms remain to be studied in depth, especially regarding whether PSP can show antidepressant-like effects by inhibiting microglia activation and regulating microglia’s M1/M2 phenotype.

Therefore, the depressive model in zebrafish was established by the chronic unpredictable mild stress for 5 weeks [15]; then, the antidepressant-like effects and the modulatory effects on microglia of PSP in CUMS-induced zebrafish were evaluated by analyzing the body weight, depressive-like behaviors in novel tank test (NTT), light and dark tank test (LDT), hypothalamic–pituitary–interrenal axis (HPI), peripheral inflammation, structure of neuron and blood–brain barrier (BBB), activation state of microglia, and M1/M2 phenotype markers. We hope this study will provide enhanced support for advancing PSP as an emotion regulation product, laying a theoretical foundation for its application in the flourishing wellness industry.

## 2. Results

### 2.1. Structural Characterization of PSP

#### 2.1.1. Chemical, Molecular Weight (Mw), and Monosaccharide Composition of PSP

As listed in Table 1, the PSP sample used in the present study was a crude polysaccharide. It contained 72.12% of total carbohydrates, 14.40% of protein, 5.68% of ash, 3.21% of water, 0.28% of flavonoids, 0.11% of phenols, and 0.03% of crude fat.

The molecular weight of the crude PSP determined by GPC/MALLS was 20.48 kDa. The GPC/MALLS profile (Figure 1B) demonstrated that PSP showed an asymmetrical broad peak. The polydispersity Mw/Mn 9.394 (Table 1) also indicated that PSP was a relatively wide distribution sample.

IC analysis (Figure 1D) demonstrated that PSP was composed of Glc, Man, and Gal, with a relative molar ratio of 98.47:1.07:0.45. The molar ratio of the three types of monosaccharides indicated that Glc was the major component of PSP.

#### 2.1.2. UV and FT-IR Analysis

The weak absorption peak at wavelength 280 nm in the UV spectrum (Figure 1A) confirmed that the PSP sample contained protein and might be a glycoprotein, which was in accordance with the protein content. The absence of an absorption peak at wavelength 260 nm confirmed that the PSP sample was devoid of nucleic acid.

The IR spectrum of PSP is shown in Figure 1C. The sample showed typical absorption peaks of polysaccharides at 3378 cm^−1^ and 2930 cm^−1^, which were attributed to the O–H and C–H stretching vibration, respectively. The absorption peaks at 1629 cm^−1^ corresponded to the asymmetric stretching of the carboxylate anions (COO–). The absorption peaks, at approximately 1420 cm^−1^, were attributed to the symmetric stretching of carboxylate anions (COO–) [16]. The pyranose ring could account for 1146 cm^−1^ and 1024 cm^−1^ signals, which corresponded to the C–O–C stretching vibration and were the typical signals of glucan. The peak at 930 cm^−1^ was ascribed to the presence of a β-glucoside bond. The absorption around 822 cm^−1^ and 771 cm^−1^ implied the existence of α-hexapyranose [17]. The lack of an absorption peak at 1740 cm^−1^ suggested that PSP did not contain uronic acid [18].

#### 2.1.3. Conformational Structure Analysis of PSP

SEM and AFM methods were applied to study the conformational morphological properties of PSP. The SEM images of PSP are shown in Figure 1E. PSP samples showed a tight structure and particle accumulation (×500). The surface texture of PSP was smooth with various-sized holes (×2000). It could be observed clearly that the surface structure was compact and full of snowflake particles when the image was enlarged to ×10,000. The polysaccharide’s internal voids contribute to its good water-solubility and enable it to engage with cell receptors, which allows it to exhibit more biological functions [19].

AFM is a valuable tool for the determination of morphological characteristics and molecular parameters of polysaccharides. The AFM planar and 3-dimensional images of the PSP are shown in Figure 1F. PSP exhibited a rough surface, compact and asymmetric shape with a height ranging from −1 to 4 nm, which is much higher than that of single chain polysaccharides (generally 0.1–1 nm), indicating that aggregations and inter and intra-molecular interactions occurred in PSP [19]. The conformation of polysaccharides is closely correlated with their biological activities.

### 2.2. Effects of PSP on the Body Weight and Depressive-like Behaviors of CUMS-Induced Zebrafish

Firstly, we performed the NTT to determine the intervention doses of PSP in zebrafish. In the NTT, no significant changes in zebrafish behavior were seen in the PSP 50, 100, 200, and 400 mg/L groups. However, the latency to the top of the zebrafish in the PSP 200 mg/L group showed a decreased tendency, and the number of top entries and the time at the top showed an increased tendency. The exploratory behavior of zebrafish in the PSP 800 mg/L group was significantly inhibited (Figure 2), with significantly longer latency to top (*p* < 0.01), significantly fewer number to top (*p* < 0.01), and significantly shorter time in top (*p* < 0.05), indicating that higher doses of PSP would have adverse effects on zebrafish. Based on the behavioral results, PSP 100 and 200 mg/L, which had positive effects on zebrafish behavior and had no toxic side effects, were selected as the intervention doses for subsequent experiments.

As shown in Figure 3E, the initial weight did not significantly differ among groups. After five weeks of CUMS, the weight of zebrafish in the CUMS group was markedly reduced (*p* < 0.05). However, the decreased body weight was improved after FLU (0.1 mg/mL) and PSP (100 and 200 mg/L) treatments (*p* < 0.05). Fluoxetine, a chronic antidepressant, like other selective serotonin reuptake inhibitors, SSRIs, is commonly utilized in experimental/rodent models of stress to validate/rescue the evoked affective states. Chronic treatment of fish with an antidepressant fluoxetine (0.1 mg/L for 8 days) normalized their behavioral and endocrine phenotypes and corrected stress-elevated IL-1β and IL-6 levels, similar to clinical and rodent data [15].

NTT and LDT were both commonly used for evaluating the depressive-like behaviors in zebrafish. Firstly, the CUMS program significantly inhibited the upward exploration behaviors of zebrafish in the NTT, manifested as the increased latency to the top (Figure 3B, *p* < 0.001), the reduced number of top entries, and the decreased time in the top (Figure 3C,D, *p* < 0.001). PSP (100 and 200 mg/L) and FLU (0.1 mg/L) treatment markedly shortened the latency to the top (Figure 3B, *p* < 0.01, *p* < 0.001) and increased the number of top entries and the time at the top (Figure 3C,D, *p* < 0.05, *p* < 0.01, *p* < 0.001). Then, in the LDT, the CUMS program significantly inhibited the bright area exploration behaviors of zebrafish, manifested as the reduced number of light entries and the decreased time in the light area (Figure 3F,G, *p* < 0.001). PSP (100 and 200 mg/L) significantly elevated the number of light entries (Figure 3F, *p* < 0.05, *p* < 0.01). The time in light area markedly increased after PSP (100 and 200 mg/L) and FLU (0.1 mg/L) treatment (Figure 3G, *p* < 0.01, *p* < 0.001). The above results indicated the success of the CUMS model in zebrafish and provided evidence that PSP could significantly ameliorate the depressive-like behaviors in CUMS-induced zebrafish.

### 2.3. Effects of PSP on the HPI Function of CUMS-Induced Zebrafish

Increased CORT levels suggested hyperactivity of the HPI axis. The results showed that the CORT levels in zebrafish body samples significantly increased after CUMS stimulation (Figure 4A, *p* < 0.001). PSP (100 and 200 mg/L) and FLU (0.1 mg/L) intervention could regulate HPI axis dysfunction because decreased CORT levels in PSP and FLU groups were observed (Figure 4A, *p* < 0.01).

### 2.4. Effects of PSP on the Peripheral Inflammation of CUMS-Induced Zebrafish

The TNF-α, IL-6, and IL-1β levels elevated, and the IL-10 levels decreased in the body samples after CUMS exposure (Figure 4B–E, *p* < 0.01, *p* < 0.001). However, the TNF-α levels were significantly reduced by PSP (200 mg/L) and FLU (0.1 mg/L) administration (Figure 4B, *p* < 0.05, *p* < 0.01), and the IL-6 and IL-1β levels were markedly reduced by PSP (100 and 200 mg/L) and FLU (0.1 mg/L) intervention (Figure 4C,D, *p* < 0.01, *p* < 0.001). Administration with PSP (200 mg/L) and FLU (0.1 mg/L) significantly increased the levels of IL-10 (Figure 4E, *p* < 0.05, *p* < 0.01). These results suggested that PSP could suppress peripheral inflammation.

### 2.5. Effects of PSP on Brain Health in CUMS-Induced Zebrafish

Nissl staining was conducted to explore the effects of PSP on neuronal damage in the mesencephalon of CUMS-induced zebrafish. Figure 5A shows that neurons were regularly arrayed, and Nissl bodies were clear in the CON group. In contrast, Nissl body disintegration and irregular and sparse neuronal arrangement were clearly observed in CUMS-induced zebrafish, and the IOD value was significantly reduced (*p* < 0.001). The decrease in Nissl-positive neurons and IOD value was significantly reversed after PSP and FLU administrations (*p* < 0.001).

The structure of neurons of zebrafish mesencephalon and telencephalon were observed using TEM (Figure 5B,C). Significant perinuclear gap expansion was observed in both mesencephalon and telencephalon of zebrafish after CUMS exposure. The mitochondrial structure changed obviously after CUMS exposure, especially in the mesencephalon, including mitochondrial swelling and loss of mitochondrial cristae in neurons. Neuronal damage in the mesencephalon was more severe than in the telencephalon region. Furthermore, the ultrastructure of the blood–brain barrier in telencephalons was observed using TEM (Figure 6). CUMS exposure induced significant basement membrane shrinkage, disrupting the integrity of the BBB. Compared with the CUMS group, PSP and FLU treatments alleviated these pathological changes. The damage to the integrity of the BBB was alleviated. The black arrow in Figure 6 indicates basement membrane shrinkage and disruption of the BBB integrity.

Altogether, these findings implied that PSP could attenuate neuronal and BBB damage induced by CUMS-induced depression in zebrafish.

### 2.6. Effects of PSP on Microglia of CUMS-Induced Zebrafish

Iba-1 is a marker protein of microglia. Its expression was analyzed by immunohistochemistry to evaluate the activation of microglia. Immunohistochemical (Figure 7) results indicated that the proportion of positive area in the mesencephalon of CUMS-induced zebrafish was markedly higher than the CON group (*p* < 0.001), indicating that microglia was activated after CUMS exposure. The activation of microglia was inhibited by PSP and FLU with a significantly reduced proportion of positive area (*p* < 0.001).

Then, to evaluate the effect of PSP on the state transition of microglia, we measured the expression of M1/M2 phenotype markers in the brain of zebrafish. As shown in Figure 8A–J, CUMS significantly increased the expression of M1 phenotype genes (*il-6*, *il-1β*, *inos,* and *cd11b*) and decreased the expression of M2 phenotype genes (*cd206* and *ym1*). The mRNA expression of M2 phenotype genes in the CUMS group (*il-4*, *il-10*, *arg-1,* and *egr-2*) showed a decreased tendency but no significant changes. These results implied that CUMS stimulation could induce phenotypic changes in microglia. However, when the zebrafish were treated with PSP, these changes were notably reversed compared to the CUMS group. PSP significantly decreased CUMS induced up-regulation of M1 phenotype genes expression (*il-6*, *il-1β*, *inos* and *cd11b*) while significantly increased CUMS induced down-regulation of M2 phenotype genes expression (*il-4*, *il-10*, *cd206*, *egr-2,* and *ym1*). The expression of *arg-1* was increased but did not show significant differences. The above results indicated that PSP could suppress neuroinflammation and modulate microglia M1/M2 polarization to exert antidepressant-like effects.

Taken togther, the antidepressant-like effects of PSP were related to altering the HPI axis hyperactivation, suppressing peripheral inflammation, inhibiting neuroinflammation induced by microglia hyperactivation, and modulating microglial M1/M2 polarization (Figure 9).

## 3. Discussion

Long-term chronic stress stimulation is an important cause of depression. CUMS was initially a procedure designed for rats or mice to induce the physiological symptoms of clinical depression by repeatedly and irregularly using a series of unpredictable mild stimuli, which now has become widely accepted as a classic procedure [20]. In rodent CUMS procedures, a series of unpredictable mild stimuli are often used, such as food and water deprivation, forced swimming, restraint stress, soiled cage, overnight illumination, and so on [21]. Then, the body weight, locomotor activities, and the classic behavioral paradigms were performed to evaluate the fundamental state of mice, exploration abilities, and depressive-like behaviors [22,23]. Referring to the rodent CUMS model, the researchers successfully establish a similar depression model in zebrafish using chronic and unpredictable stress, which can also induce strong behavioral and physiological changes in zebrafish. The main stressors currently used on zebrafish are predator exposure, light-dark exposure, net chasing, crowding, shallow water exposure, social isolation, alarm pheromones, air exposure, and tri-colored cups [15]. Several behavioral tests have been developed to assess depression in zebrafish, among which the NTT and LDT are widely used. The NTT can evaluate the depressive-like behavior of zebrafish from the overall movement, upward exploration, and freezing. Zebrafish have a preference for darkness; in the LDT, the decreased time in the light area corresponded to the enhancement of anxiety and depression [24]. In addition, consistent with rodents, CUMS-induced zebrafish also experience malnutrition and weight loss [25].

In the present study, the intervention doses of PSP were first determined through the NTT. PSP intervention has a positive impact on zebrafish behavior within a certain dose range. However, excessive intervention doses actually inhibit zebrafish exploratory behavior. We speculate that excessive doses of PSP are toxic to zebrafish. Therefore, PSP (100 and 200 mg/L), which had positive effects on zebrafish behavior and had no toxic side effects, was selected as the intervention concentration for the subsequent experiments. The body weight of zebrafish was significantly reduced after five weeks of CUMS procedure. The exploratory behaviors of CUMS-induced zebrafish in NTT and LDT were significantly inhibited, demonstrating the success of the CUMS-induced depression model in zebrafish. A significant reversal of these behavioral changes was observed in CUMS-induced zebrafish after administering PSP. The notable antidepressant-like effects of PSP were consistent with the previous studies in mice [13].

Depression patients not only exhibit behavioral and emotional disorders but are also accompanied by changes in brain structure and dysfunction. In CUMS model of the rodents, neuronal damage characterized by brain tissue atrophy, neuron loss, and apoptosis can also be observed [26]. The present study evaluated the effects of CUMS and PSP on zebrafish brain neurons through Nissl staining and TEM observation. Nissl staining can reflect neurons’ survival and neuroplasticity levels to a certain extent [27]. The results showed that neurons were irregularly distributed as well as Nissl-bodied disintegrated, and the IOD value significantly decreased in the mesencephalon of CUMS-induced zebrafish, which indicated that the neurons were obviously damaged after CUMS exposure. TEM is used to observe the ultrastructure of neurons [28]. Significant perinuclear gap expansion in both mesencephalon and telencephalon and mitochondrial structure damage in mesencephalon were observed in zebrafish after CUMS exposure. PSP significantly alleviated neuronal damage in zebrafish with the increased Nissl bodies, reduced perinuclear expansion, and restored mitochondrial structure. BBB is a crucial structure to maintain the proper function of the brain. Disruption of the BBB is observed in several different neurological disorders, including depression. Research showed that targeted disruption of the female BBB induced anxiety and depression-like behaviors [29]. Recent studies implicate that inflammation may be one of the mechanisms of BBB disruption [30]. In this study, PSP partially reversed the BBB disruption, which may be related to its anti-inflammatory effects, and further research is needed afterward.

The HPA axis can effectively regulate the physiological response to stress, and the HPA axis hyperfunction is a typical feature of depression patients and depression model animals. In CUMS model of rodents, a high level of glucocorticoid produces a range of neurotoxic effects in the prefrontal cortex and hippocampus, inducing microglia activation, neuronal apoptosis, neuroplasticity injury, and peripheral and neuroinflammation [31]. The HPI axis of zebrafish is structurally and functionally homologous to the HPA axis of mammals. The HPI axis, as the pressure regulation system of fish, plays an important role in maintaining the fish’s stable status under various physiological or environmental pressures [32]. In zebrafish, cortisol is the major product of the HPI axis, so it is often used as a stress response indicator [32]. When faced with CUMS stimulation, the CORT levels in the body of zebrafish significantly increased. However, in the present study, it was reversed after PSP and FLU treatment, which was in agreement with the previous studies in mice, suggesting that adjustment of the HPA axis might be one of the mechanisms of PSP antidepressant effects [12]. The functional study of the HPI axis in fish also involves some important receptors, such as glucocorticoid receptor (GR) and mineralocorticoid receptor (MR) [33,34]. Further research is needed on the effects of CUMS and PSP on the related receptors and their role in the antidepressant effects.

The bidirectional communication between the endocrine and immune systems jointly regulates the body’s homeostasis. It is known that many cytokines interact with the HPA axis and positively correlate with the degree of depression. TNF-α, IL-1β, and IL-6 are considered to be the major participants in this bidirectional communication [35]. In addition, IL-10 is an important anti-inflammatory cytokine that can inhibit neuronal apoptosis and exert neuroprotective effects by activating the JAK1/STAT3 signaling pathway [36]. IL-10 can also inhibit the activation of microglia and the production of pro-inflammatory factors [37]. In our present study, CUMS exposure induced peripheral inflammatory response in zebrafish. The levels of pro-inflammatory cytokines were significantly increased, and the release of the anti-inflammatory cytokines decreased in the zebrafish body. In contrast, these changes were reversed after PSP administration. PSP could inhibit peripheral inflammation by regulating the balance of pro-inflammatory and anti-inflammatory factors and inhibiting the excessive activation of the HPI axis.

Peripheral inflammation and chronic stress can cause neuroinflammation related to activated microglia. Microglia is a multifunctional immune cell in the CNS. It monitors and maintains homeostasis of the internal environment through its dynamic process [38]. The transformation of microglia from a resting ramified state to an ameboid active state and functional phenotypes are the focus of attention in neuropsychiatric diseases [39]. Inhibiting the over-activation and regulating the polarization phenotype of microglia can be potential targets for the treatment of neuropsychiatric disorders. Different polarized microglia phenotypes, commonly called M1/M2 phenotypes, play destructive and protective roles in the CNS, respectively. M1 phenotype microglia release neurotoxic substances and inflammatory factors, leading to neurological dysfunction and inducing inflammatory storm, while M2 phenotype microglia restore CNS homeostasis by blocking inflammatory processes and promoting neurotrophic factor production [40]. In LPS and CUMS induced depression models of rodents, neuroinflammation accompanied by the increase in M1 microglia and the decrease in M2 microglia was observed [41]. Zebrafish microglia share a highly conserved signature gene program with their mammalian counterparts. Both also share functional similarities, including responding to danger signals, clearing apoptotic neurons, and fine-tuning neuronal activity [42]. The activation of microglia also exists in zebrafish. Microglia was activated in Tilapia Lake Virus-infected larvae. The cell shape changed from a ramified state to an ameboid active state. Strong neuroinflammation was also induced by the Tilapia Lake Virus in adult fish, and the expression of microglia genes was upregulated [43]. Administration of IGF-1 morpholino at the lesion site of spinal-transected zebrafish increased the immunofluorescence density of iba-1, suggesting that microglia were activated [44]. Chronic Piper methysticum treatment upregulated several microglial (iNOS, Egr-2, CD11b) biomarker genes [45]. The previous study identified two distinct microglial subtypes, phagocytotic and regulatory microglia, which may have complementary functions under physiological and pathological conditions in adult zebrafish. Phagocytotic microglia are amoeboid in shape, but regulatory microglia have ramified protrusions. The former is highly mobile and phagocytotic, while the latter is the opposite [42]. The activation of microglia M1 phenotype (CD11b) and the downregulation of microglia anti-inflammatory phenotype (CD206) were found in crhr2-/- zebrafish [46]. In this study, the activation status and polarization phenotypes of microglia and neuroinflammation in CUMS-induced zebrafish were investigated with reference to rodent microglia marker genes. Our results suggested that CUMS exposure in zebrafish induced neuroinflammation, microglia activation and promoted M1 phenotype polarization. PSP was an important modulator of microglial inflammation by inhibiting microglia activation and promoting M2 phenotype polarization. It is of great significance to understand the role of microglia in depression and the regulation of PSP on microglia.

The efficacy of polysaccharides is closely related to their structure, including molecular weight, composition of monosaccharide, type of glycosidic bond, chemical modification, etc. Studying the structure-activity relationship of polysaccharides is of great significance for in-depth research on biological activity and its development and utilization [47]. Most polysaccharides having antidepressant-like effects are heteropolysaccharides composed of two or more monosaccharides with a high proportion of Glc. Those containing a β-glycosidic bond appear to have good antidepressant activity [48]. The structure of PSP in this study conforms to the above rules. However, there is currently limited research on the relationship between the structure and antidepressant effects of polysaccharides, and further research is needed in the future.

## 4. Materials and Methods

### 4.1. Materials and Reagents

PSP was obtained from Shanghai Yuanye Bio-Technology Co., Ltd. (Shanghai, China). PSPs are aqueous extracts obtained from the PS rhizomes by water extraction and alcohol precipitation methods. Fluoxetine hydrochloride (FLU) was purchased from MedChemExpress LLC (Shanghai, China). Commercial sandwich enzyme-linked immunosorbent assay (ELISA) kits for cortisol (CORT), tumor necrosis factor (TNF-α), interleukin (IL)-6, IL-1β, and IL-10 were obtained from Jianglai Biological Technology Co., Ltd. (Shanghai, China). qRT-PCR primers were obtained from Tsingke Biotechnology Co., Ltd. (Beijing, China). Toluidine blue O was obtained from Wuhan Servicebio Technology Co., Ltd. (Wuhan, China). A DAB kit was obtained from Epsilon Biotechnology Co., Ltd. (Beijing, China). All other reagents were of analytical grade.

### 4.2. Structure Characterization of PSP

#### 4.2.1. Determination of Content of Total Carbohydrate, Protein, Phenols, Flavonoids, Crude Fat, Ash and Water

The contents of total carbohydrates, protein, phenols, flavonoids, crude fat, ash were quantified by the phenol–sulfuric acid method [49], the bicinchoninic acid method [50], the Folin–Ciocalteu method, the NaNO_2_-Al(NO_3_)_3_-NaOH colorimetric method [51], the Soxhlet extraction method and carbonization method [52], respectively. The water content was measured using a moisture-quick tester (Mettler Toledo MJ33, State of Delaware, USA).

The total sugar content was determined using the Grace Polysaccharide Content Kit manufactured by Grace Biotechnology (Suzhou, China) according to the phenol–sulfuric acid method. The sample was supplemented with a total of 3 mg. Additionally, 2 mL of distilled water was added. The mixture was then heated in a boiling water bath for 2 h, shaken, and mixed a few times every 20 min. After heating, the mixture was removed and placed at room temperature, with a rotational speed of 8000 rpm for 5 min. The 200 μL supernatant was transferred to a new EP tube with 1 mL of ethanol and mixed well at 4 °C for 1 h, with a rotational speed of 8000 rpm for 5 min. A total of 1 mL of 80% ethanol was added to the precipitate and mixed well at 8000 rpm for 5 min before the supernatant was discarded, and the precipitate was left to avoid precipitation. A total of 2 mL of distilled water was added to the precipitate and heated in a boiling bath until completely dissolved. A 100 μL solution of polysaccharide test was added to the EP tube, along with 150 μL of reagent I and 250 μL of concentrated sulfuric acid. After mixing well, the solution was heated in a 95 °C water bath for 20 min and then cooled to room temperature. The absorbance was measured at 488 nm.

The total phenols content was determined using the Grace Total Phenols Content Kit (Folin-Ciocalteu method). A total of 0.03 g of the sample was added with 1.5 mL of 60% ethanol, shaken and extracted at 60 °C for 2 h, placed at room temperature, centrifuged at 12,000 rpm for 10 min, and the supernatant was collected for testing. Sequentially, 10 μL of samples and 50 μL of reagent I were added to the 96 well plate. The solution was mixed well at 25 °C for 3 min. A total of 50 μL of reagent II was added and mixed well at 25 °C for 6 min. Then, a total of 105 μL of reagent III was added and mixed well at 25 °C for 30 min. The absorbance was measured at 760 nm.

The total flavonoid content was determined using the Grace Total Flavonoids Content Kit (NaNO_2_-Al (NO_3_)_3_-NaOH method). A total of 0.03 g of the sample was added with 1.5 mL of 60% ethanol, shaken and extracted at 60 °C for 2 h, placed at room temperature, centrifuged at 12,000 rpm for 10 min, and the supernatant was collected for testing. A total of 50 μL of the samples and 15 μL of reagent Ι were added and sequenced to the 96 well plate and mixed well at 25 °C for 6 min. A total of 30 μL of reagent II was added and mixed well at 25 °C for 6 min. A total of 105 μL of reagent III 105 μL was added and mixed well at 25 °C for 15 min. The absorbance was measured at 510 nm.

The protein content was determined using the BCA kit. A 10 mg portion of the sample was combined with 200 μL of RIPA lysis solution and was subjected to lysis for 30 min, followed by centrifugation at 12,000 rpm for 10 min, and the supernatant was collected for testing. A BCA working solution of 200 μL was added at 37 °C for 30 min. The absorbance was measured at 562 nm.

The Soxhlet extraction method was used to determine the content of crude fat. Reflux extraction with petroleum ether was performed on 2–5 g of the sample. The extraction was conducted at a rate of 8 times/h for 8 h. When 1 mL-2 mL of solvent remained in the receiving bottle, it was removed, recovered with petroleum ether, and steam dried in a water bath. Then, it was dried at 100 °C ± 5 °C for 1 h, cooled in a dryer for 0.5 h, and weighed. The above operation was repeated until the weight was constant.

#### 4.2.2. Molecular Weight (Mw) Analysis

The molecular weight of the sample was measured using a gel permeation chromatography-eighteen-angle laser light scattering instrument (GPC-MALLS). This process was performed using the gel column (Shodex SB-806) and the combined detectors (Wyatt Dawn Heleos II eighteen-angle laser light scattering and Wyatt Optilab rEX refractive index detector). The mobile phase was a 0.1 mol/L sodium chloride solution at a flow rate of 0.5 mL/min, and the column temperature was 40 °C. The injection volume was 0.1 mL. The sample was dissolved in 0.1 mol/L sodium chloride solution at a concentration of 2 mg/mL.

#### 4.2.3. Monosaccharide Composition Determination

The polysaccharide sample (10 mg) was hydrolyzed with 4 mol/L trifluoroacetic acid at 120 °C in a sealed tube for 4 h, and the hydrolysate was dried with nitrogen to remove excess trifluoroacetic acid. The polysaccharide hydrolysate was dissolved with 10 mL distilled water and then passed through a 0.2 μm microporous filter. Chromatographic separation conditions of fucose (Fuc), rhamnose (Rha), arabinose (Ara), galactose (Gal), glucose (Glc), fructose (Fru), glucuronic acid (GlcA) and galacturonic acid (Gala): Ion Chromatography (IC, DIONEX ICS-3000, California, USA), CarboPacTMPA10 (4 × 250 mm) analytical column, using sodium acetate (1 mol/L), sodium hydroxide (200 mmol/L) and water as the mobile phase at a flow rate of 1 mL/min. Chromatographic separation conditions of xylose (Xyl) and mannose (Man): IC (DIONEX ICS-3000, California, USA), CarboPacTMPA20 (3 × 150 mm) analytical column, using sodium hydroxide (250 mmol/L) and water as the mobile phase at a flow rate of 0.5 mL/min. The column temperature was 35 °C, and a 10 μL sample was injected.

#### 4.2.4. Ultraviolet (UV) Spectrum Scan and Fourier Transform Infrared Spectroscopy (FT-IR)

The spectral data were obtained on a UV–visible spectrophotometer (Metash UV-8000 s, Shanghai, China) at 25 °C in the 200–600 nm range.

The IR spectrum was recorded on a Fourier transform infrared spectroscopy (Bruker Tensor 27, Saarbrücken, Germany) by grinding a mixture of the sample (2 mg) with dry KBr (100–200 mg for 24 h at 120 °C) and pressing in a mold. The frequency range was 4000–400 cm^−1^.

#### 4.2.5. Scanning Electron Microscope (SEM) Analysis

An aliquot of PSP was spread on a mica sheet and sputtered a 20 nm thick gold film under a vacuum. Afterward, the surface of the sample was photographed using SEM (Hitachi SU8010, Toyko, Japan).

#### 4.2.6. Atomic Force Microscope (AFM) Analysis

A total of 2.5 μL of 0.2 mg/mL PSP solution was deposited on a clean mica disk and dried at 20 °C. Then, the images were observed by AFM (Park NX10, Suwon City, Republic of Korea).

### 4.3. Animal Experiment

#### 4.3.1. Animals

Adult wild-type zebrafish (4–6 months of age) were obtained from an aquarium (Ornamental fish supplier, Beijing, China). The animals were maintained at 27 °C in filtered system water (pH = 7.0–7.6) with a light (14 h)–dark (10 h) cycle, according to standards. All animals received fresh Artemia twice daily unless indicated otherwise in the CUMS procedure. The Artemia spp. Nauplii are used as live food throughout the development of several aquatic organisms, including fish and shrimp, to improve survival rates in hatcheries [53].

Animal experimentation fully complied with National and institutional guidelines and regulations and was approved by the ethics committee for research on laboratory animal use at the institution. The study was performed under the approval and supervision of the Animal Ethics Committee at the Institute of Food Science and Technology, Chinese Academy of Agricultural Sciences (Approval Number: SYXK-2023061528).

#### 4.3.2. Evaluation of PSP after Acute Administration

Firstly, in order to determine the intervention doses of PSP in subsequent experiments, zebrafish were randomly divided into 6 groups: CON (control group), PSP (50 mg/L), PSP (100 mg/L), PSP (200 mg/L), PSP (400 mg/L) and PSP (800 mg/L). Zebrafish were housed in groups of 15 per tank in 10 L tanks. PSP was dissolved in another container with filtered system water. After feeding in the morning, the zebrafish were transferred to the container of PSP with the corresponding concentration and immersed for 1 h, continuously immersing for 9 days. Zebrafish in the CON group were not subjected to any administration. The novel tank test (NTT) was performed on days 8 and 9.

#### 4.3.3. CUMS Procedure and PSP Administration

The CUMS procedure was performed as previously described with appropriate modification [24]. The following stimuli were selected for study: predator exposure, predator water, net chasing, crowding, darkness, light-dark exposure, super-bright light, social isolation, red bucket, tri-colored cups, food deprivation, shallow water exposure, alarm pheromone, air exposure, shaking. Zebrafish were continuously exposed to several unpredictable stressors for 5 weeks. The zebrafish in the control group were separated from the stressed groups.

The animals were given 14 days to acclimation and then divided into 5 groups: CON (not subjected to any stress), CUMS (CUMS procedure), FLU (FLU 0.1 mg/mL treatment + CUMS), PSP-L (PSP 100 mg/L + CUMS), PSP-H (PSP 200 mg/L + CUMS). Zebrafish were housed in groups of 30 per tank in 10 L tanks. Except for the CON group, the CUMS, FLU, PSP-L, and PSP-H groups were subjected to CUMS procedures after feeding in the morning for 5 weeks. Starting from the 4th week of CUMS procedure, the zebrafish were transferred to the container of FLU and PSP with the corresponding concentration and immersed for 1 h. FLU and PSP treatment continuously for 9 days (days 29–38) before the beginning of the daily CUMS procedure and behavior tests. Fish in the CON group and CUMS group were not subjected to any PSP and FLU treatment. The body weight of zebrafish was monitored (day 35), and NTT and the light and dark tank test (LDT) were performed after the CUMS procedure and administration (days 35–38). The schedule of the animal experiment is exhibited in Figure 2A.

### 4.4. Behavioral Tests

#### 4.4.1. Body Weight

The body weight of zebrafish was monitored on days 0 and 35. The weight of the beaker with water was measured. Then, the zebrafish was fished out with a dry net and quickly put into the measuring beaker, calculating the weight difference before and after. The excess water needs to be drained during the transfer process.

#### 4.4.2. Novel Tank Test (NTT)

The novel tank test was conducted as previously described [54]. NTT was performed between 9:00 a.m. and 6:00 p.m. All zebrafish fasted for 24 h before NTT. The apparatus was a 2.5 L rectangular tank (25 cm length × 5 cm width × 20 cm height) and maximally filled with water. It was divided into two equal virtual horizontal portions by a line marking the outside walls. A single zebrafish in the tank was recorded for 5 min with a high-definition camera to manually score the latency to top (time taken to cross the line for the first time, s), time spent in top (s), top entries (the number of transitions to top).

#### 4.4.3. Light and Dark Tank Test (LDT)

The light and dark tank test was performed according to a previous report [55]. LDT was performed between 9:00 a.m. and 6:00 p.m. All zebrafish fasted for 24 h before LDT. The apparatus was a 10 L rectangular tank (28 cm length × 20 cm width × 18 cm height) and maximally filled with water. It was divided into two equal portions: dark and light. A partition could be inserted between the bright and dark areas. The zebrafish could move when the partition is removed. A single zebrafish was placed in the dark areas with a partition inserted to acclimation for 2 min and recorded for 5 min with a high-definition camera after removing the partition to manually score the time spent in light (s), light entries (the number of transitions to light area).

### 4.5. Sample Collection

All zebrafish were immediately sacrificed in an ice water bath after behavior tests. The head and body of zebrafish were separated on the ice. The whole-brain samples were collected for gene expression studies and quickly frozen in liquid nitrogen. Mesencephalic and telencephalic regions were collected in corresponding fixed solutions for further pathological and morphological analyses. The headless whole-body samples of zebrafish were collected and quickly frozen in liquid nitrogen. The whole-brain and whole-body samples were stored at −80 °C for further assays.

### 4.6. Nissl Staining

After sacrifice, the mesencephalon samples were paraffin-embedded after being fixed with 4% paraformaldehyde. Then, the paraffin sections (4 μm) were dewaxed and rehydrated. The sections were incubated with toluidine blue for 5 min at room temperature. The sections were then differentiated with 0.1% glacial acetic acid for several seconds, followed by rinsing in double distilled water cleared in xylene, covered by neutral resins, and examined under a microscope (Nikon DS-U3, Toyko, Japan). The integrated optical density (IOD) was counted with Image J software 1.8.0 (National Institutes of Health, Bethesda, MD, USA).

### 4.7. Transmission Electron Microscope (TEM) Observation

After sacrifice, the telencephalon and mesencephalon samples were quickly divided into small pieces on ice and fixed in 2% glutaraldehyde. Post-fixation, samples were immersed in 1% osmium tetraoxide for 2 h at 4 °C and then dehydrated using 50, 70, 90 and 100% alcohol. Samples were embedded in Epon 812, and ultrathin sections (40–50 nm) were contrasted with uranyl acetate followed by lead citrate and examined with TEM (Hitachi H-7500, Toyko, Japan) [56].

### 4.8. Immunohistochemistry

The immunohistochemical analysis was carried out according to the previous study [57]. Samples of mesencephalon are fixed, embedded, and sectioned in the same manner as Nissl staining. After retrieving the antigen with citrate buffer (pH = 6.0) and quenching endogenous peroxidase activity with 0.3% H_2_O_2_, the sections were blocked with 3% BSA. Then, the sections were incubated overnight at 4 °C with Iba-1 antibody (ab178847, Abcam), washed and incubated with secondary antibodies for 1 h at room temperature, and stained with DAB. The sections were counterstained, dehydrated, and examined under a microscope (Nikon DS-U3, Toyko, Japan). The proportion of positive areas was counted with Image J software.

### 4.9. RNA Isolation and Real-Time Quantitative PCR

Three zebrafish whole brains were used to prepare a single sample. RNA was isolated according to the EASYspinPlus rapid tissue/cellular RNA extraction kit (Aidlab Biotechnologies, Beijing, China). GoScript Reverse Transcription System (Promega Biotech, Beijing, China) was used for the synthesis of cDNA. qRT-PCR was performed in Applied Biosystems QuantStudio 7 Flex system (Thermo FisherScientific, Waltham, MA, USA) using NovoStart SYBR qPCR SuperMix Plus kit (Novoprotein, Beijing, China). Relative gene expression was calculated using the 2^−ΔΔCT^ method and normalized with control gene β-actin. Primer information is shown in Table 2.

### 4.10. Enzyme-Linked Immunosorbent Assay (ELISA)

The headless whole-body samples were used to measure CORT, TNF-α, IL-1β, IL-6, and IL-10 levels using commercial ELISA kits (Jianglai Biological Technology, Shanghai, China). Briefly, the headless body sample of a single zebrafish was cut into small pieces on ice and homogenized with an electric grinder in 1 mL PBS, centrifuged at 10,000× *g* for 10 min, 4 °C. The supernatant was collected and tested based on the manufacturer’s instructions.

### 4.11. Statistical Analysis

Software used for the statistical analysis included SPSS 23.0, GraphPad Prism 8.0, and Image J. The results were expressed as mean ± standard error mean (Mean ± SEM). The homogeneity and normality were checked, and then, a one-way ANOVA followed by Fisher’s least significant difference (LSD) test was applied to compare the differences among groups. *p* < 0.05 was considered statistically significant.

## 5. Conclusions

PSP samples had the typical structure characteristics of polysaccharides, consisting of Glc, Man, and Gal, with an average Mw 20.48 kDa. Conformational structure results showed that PSP owned a porous and agglomerated morphology. PSP was demonstrated to have remarkable effectiveness in reversing depressive-like behaviors in CUMS-induced zebrafish depression model. It altered the HPI axis hyperactivation and alleviated neuronal and blood–brain barrier damage in the mesencephalon and telencephalon of CUMS-zebrafish. Furthermore, it was confirmed that the antidepressant effects of PSP have a potential association with microglia. On the one hand, PSP inhibits microglial polarization. On the other hand, PSP inhibits inflammatory response by regulating the transformation of microglia into M2 phenotype (Figure 9). The underlying mechanisms of PSP in emotion regulation, especially depression, remain a subject of ongoing exploration and inquiry. The findings of this study contribute to a deeper understanding, which provides valuable insights for further refinements and enhancements.

## Figures and Tables

**Figure 1 ijms-25-02005-f001:**
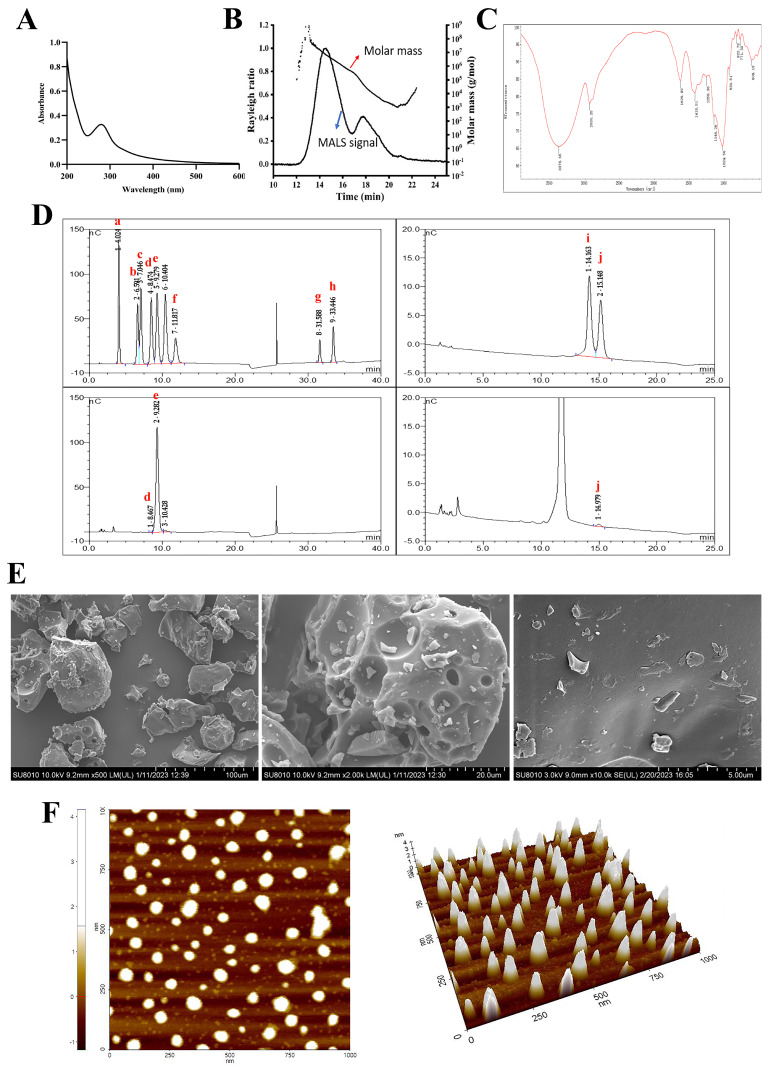
Structure characterization of PSP. (**A**) Ultraviolet spectrum scan, (**B**) molecular weight, (**C**) Fourier transform infrared spectroscopy, (**D**) monosaccharide composition, (**E**) scanning electron microscope, and (**F**) atomic force microscope results of PSP. (a. Fucose; b. Rhamnose; c. Arabinose; d. Galactose; e. Glucose; f. Fructose; g. Glucuronic acid; h. Galacturonic acid; i. Xylose; j. Mannose).

**Figure 2 ijms-25-02005-f002:**
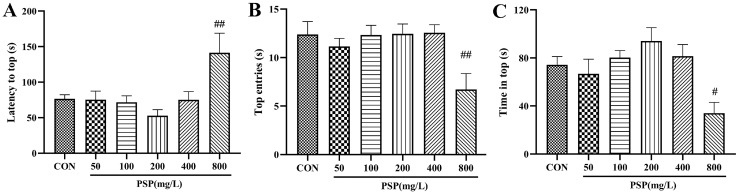
Determination of PSP intervention doses in the novel tank test (NTT). (**A**) Latency to top. (**B**) Top entries. (**C**) Time in the top. (*n* = 12–15). # *p* < 0.05, ## *p* < 0.01 compared with the CON group.

**Figure 3 ijms-25-02005-f003:**
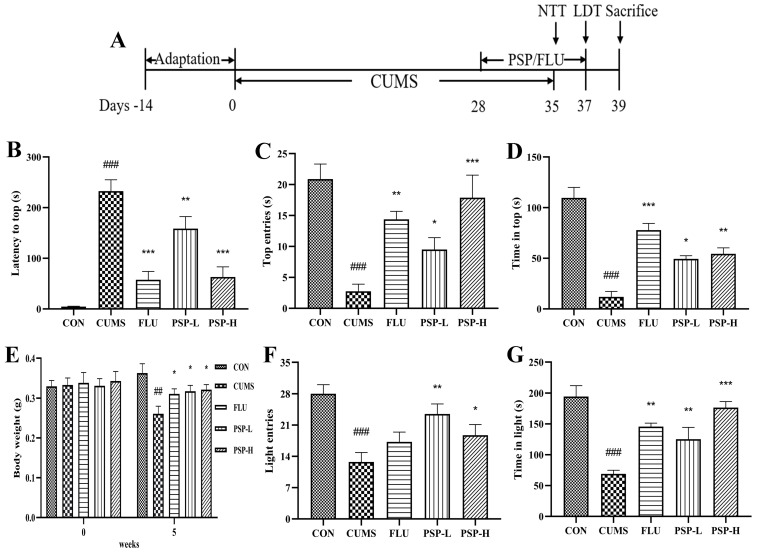
Effects of PSP on the body weight and depressive-like behaviors in CUMS-induced zebrafish. (**A**) Schedule of the experimental procedure. (**B**) Latency to top, (**C**) top entries, and (**D**) time in top in the novel tank test (NTT). (**E**) Changes in the body weight. (**F**) Light entries and (**G**) time in light in the light and dark tank test (LDT). (*n* = 8). ## *p* < 0.01, ### *p* < 0.001 compared with the CON group, * *p* < 0.05, ** *p* < 0.01, *** *p* < 0.001 compared with the CUMS group.

**Figure 4 ijms-25-02005-f004:**
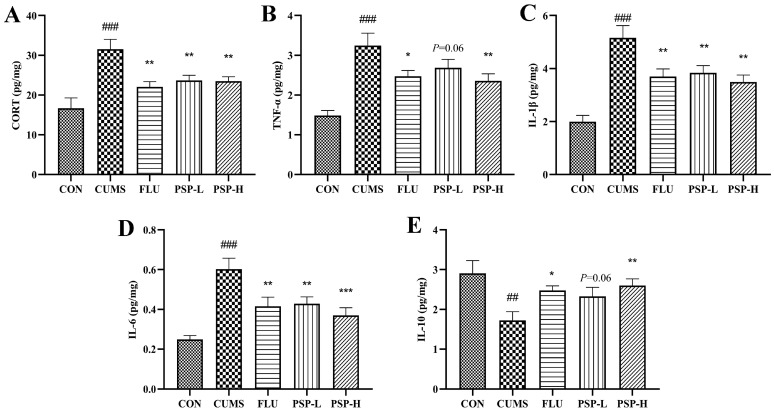
Effects of PSP on the HPI axis and peripheral inflammation in CUMS-induced zebrafish. (**A**) The cortisol (CORT) levels, (**B**) tumor necrosis factor (TNF-α) levels, (**C**) interleukin (IL)-1β levels, (**D**) IL-6 levels, and (**E**) IL-10 levels in body samples. (*n* = 8). ## *p* < 0.01, ### *p* < 0.001 compared with the CON group, * *p* < 0.05, ** *p* < 0.01, *** *p* < 0.001 compared with the CUMS group.

**Figure 5 ijms-25-02005-f005:**
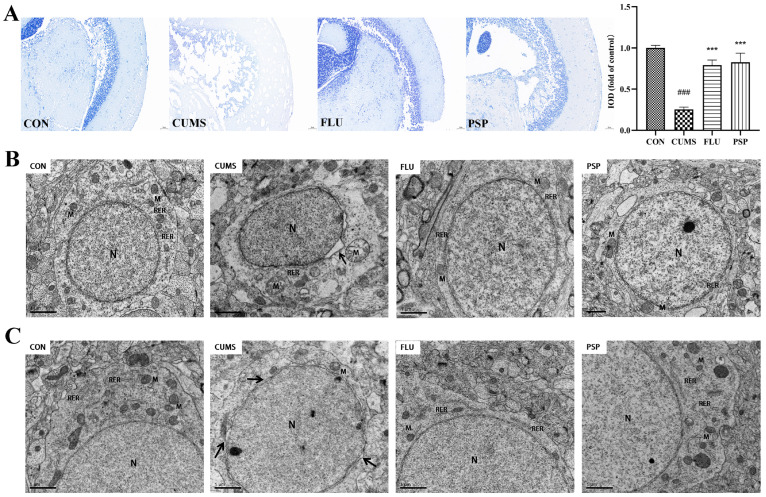
Results of Nissl staining and neuronal ultrastructure of zebrafish in each group. (**A**) Representative Nissl staining images of zebrafish mesencephalon in the CON, CUMS, FLU, and PSP-H group Scale bar = 50 μm. Neuronal ultrastructure of zebrafish (**B**) mesencephalon and (**C**) telencephalon in the CON, CUMS, FLU, and PSP-H group. (N, the nucleus, M, mitochondria, RER, rough endoplasmic reticulum). Black arrow, perinuclear gap expansion. Scale bar = 1 μm. ### *p* < 0.001 compared with the CON group, *** *p* < 0.001 compared with the CUMS group.

**Figure 6 ijms-25-02005-f006:**
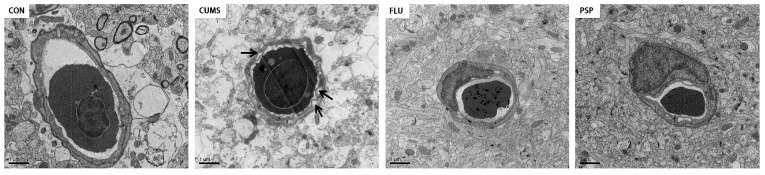
Blood–brain barrier ultrastructure of zebrafish telencephalon in the CON, CUMS, FLU, and PSP-H group. Black arrow, basement membrane shrinkage and disruption of the BBB integrity. Scale bar = 1 μm.

**Figure 7 ijms-25-02005-f007:**
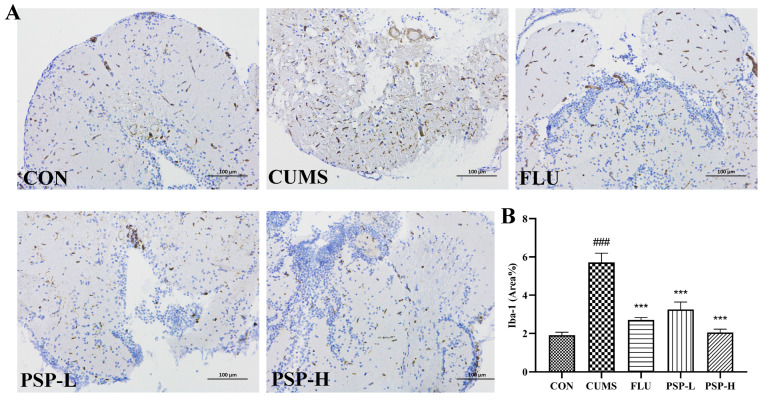
Effects of PSP on the expression of Iba-1 in CUMS-induced zebrafish. (**A**) Representative immunohistochemical images of Iba-1 in the CON, CUMS, FLU, PSP-L, and PSP-H group. (**B**) Results of the proportion of Iba-1 positive areas. (*n* = 4). Scale bar = 100 μm. ### *p* < 0.001 compared with the CON group, *** *p* < 0.001 compared with the CUMS group.

**Figure 8 ijms-25-02005-f008:**
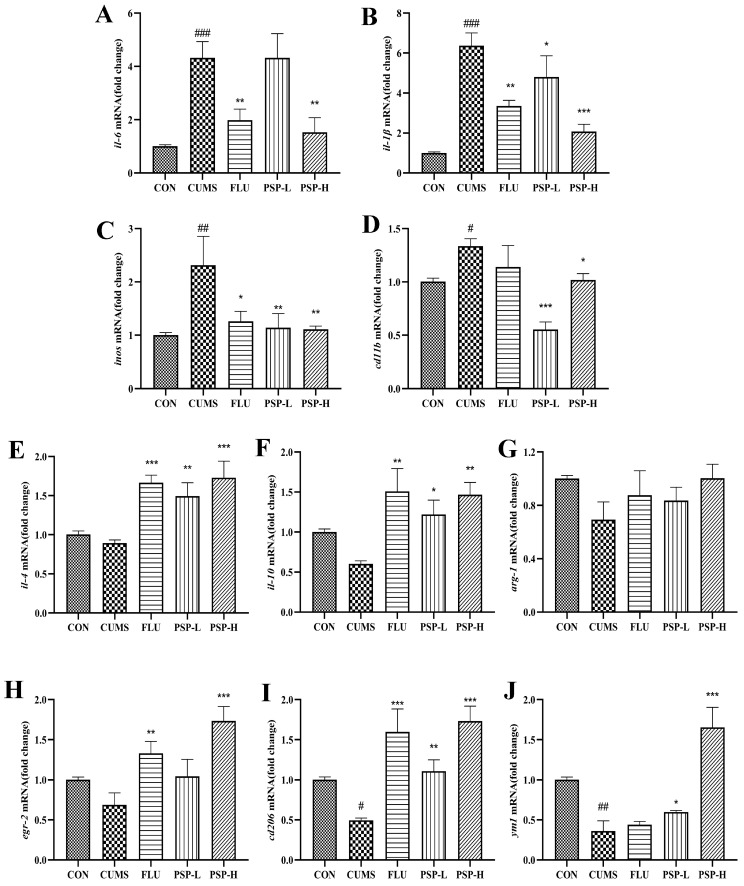
Effects of PSP on mRNA levels of M1/M2 microglial markers in CUMS-induced zebrafish. The mRNA expression of (**A**) *il-6*, (**B**) *il-1β*, (**C**) *inos*, (**D**) *cd11b*, (**E**) *il-4*, (**F**) *il-10*, (**G**) *arg-1*, (**H**) *egr-2*, (**I**) *cd206* and (**J**) *ym1*. (*n* = 6). # *p* < 0.05, ## *p* < 0.01, ### *p* < 0.001 compared with the CON group, * *p* < 0.05, ** *p* < 0.01, *** *p* < 0.001 compared with the CUMS group.

**Figure 9 ijms-25-02005-f009:**
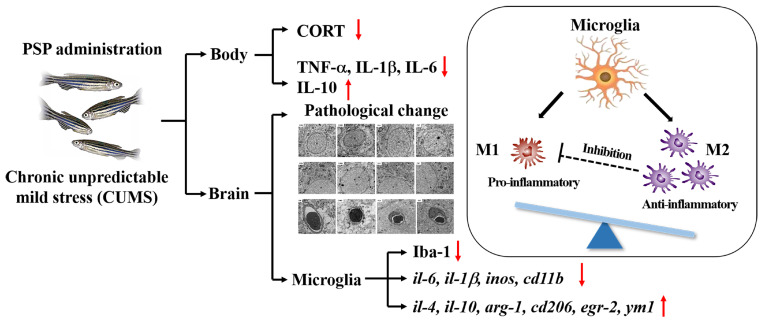
The mechanisms by which PSP exerts antidepressant-like effects in CUMS-induced zebrafish.

**Table 1 ijms-25-02005-t001:** The chemical, molecular weight (Mw), and monosaccharide composition of PSP. (Mean ± SEM).

Parameters	Units
Molar mass moments (g/mol)	
Weight-average molecular weight (Mw)	2.048 × 10^4^ ± 0.03
Number-average molecular weight (Mn)	2.180 × 10^3^ ± 0.05
Peak-position molecular weight (Mp)	1.557 × 10^3^ ± 0.04
Z-average molecular weight (Mz)	7.340 × 10^6^ ± 0.08
Polydispersity	
Mw/Mn	9.394 ± 0.08
Chemical composition (%, *w*/*w*)	
Total carbohydrate	72.12 ± 0.90
Protein	14.40 ± 0.004
Ash	5.68 ± 0.51
Water	3.21 ± 0.08
Flavonoids	0.28 ± 0.06
Phenols	0.11 ± 0.003
Crude fat	0.03 ± 0.008

**Table 2 ijms-25-02005-t002:** Primer sequences used in real-time quantitative PCR assay.

Gene	Forward Primer	Reverse Primer
*β-actin*	ACCACGGCCGAAAGAGAAAT	ATGTCCACGTCGCACTTCAT
*inos*	CCTCCTCATGTACCTGAATCTCG	GCTCCTTGCTTTAGTATGTCGC
*cd11b*	TCCTCGGATTCCAGAAACAC	AGCAGCACAAGTCCTCCAAT
*ym1*	GCAAGAGGAAGTCCACCTGAAGAC	ATACAGCAGCGGTCAGCATAAGC
*arg-1*	TCCGTTCTCCAAAGGACAGC	GACTCGTCGTTGGGAAGGTT
*cd206*	ACGCTTTCGATGGGTTTCCT	CCCTCCGTAGTACATTCCGC
*egr-2*	TCTGGATGCGGAGAGGTCTATCAAG	AGTAGGATGGCGGAGGATATGAGATG
*il-4*	TTGGTCCCCGTTTCTGAGTC	CCAGTCCCGGTATATGCTGC
*il-10*	AAGCACTCCACAACCCCAAT	TGCATTTCACCATATCCCGCT
*il-6*	AGACCGCTGCCTGTCTAAAA	TTTGATGTCGTTCACCAGGA
*il-1β*	TTCCCCAAGTGCTGCTTATT	AAGTTAAAACCGCTGTGGTCA

## Data Availability

The data supporting this study’s findings are available from the corresponding author upon request.

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
