# Peer review of "Structural Characterization and Antidepressant-like Effects of Polygonum sibiricum Polysaccharides on Regulating Microglial Polarization in Chronic Unpredictable Mild Stress-Induced Zebrafish"

_ijms, 2024, doi:10.3390/ijms25042005_

Round 1

Reviewer 1 Report

Comments and Suggestions for Authors

The paper by Zhang et al. " Structural characterization and antidepressant-like effects of Polygonum sibiricum polysaccharides associated with regulating microglial polarization in CUMS-induced zebrafish covered an interesting and actual field of research.

The study attempts to dissect the mechanism below the anti-depressant effects of the Polygonum sibiricum polysaccharides on the Zebrafish model.

This could be an interesting paper; however, I strongly suggest that the authors should significantly improve their manuscript.

Main concerns

The title contains an acronymus: “CUMS” that cannot be explained in 'this seat' but that is also never explained in the text. It is suggested to replace it in the title with 'stress' and to give a definition in the text.

The Abstract should be improved, it is not well written and a revision of English and of its construction is necessary.

Line 43: Why do you limit the context to healthy to China???

Line 48. Please, add references to the sentence: “Neuroinflammation is considered as a pathological mechanism of depression. “

Line 57: please, add more references that were already available well before the only authors mentioned.

Line 58-60: please, add references.

Line 61-63: Since minor toxic and adverse effects have been mentioned, please add references.

Line 73: please add reference for “chronic unpredictable mild stress “ adopted .

Line 76: please add the references for novel tank test (NTT), light and dark tank test (LDT) and spend some words to explain how these tests work.

Line 79-81: Rephrase the sentence (more scientifically).

Lines 85-90. It is suggested to postpone this part at the beginning of paragraph 2.3.

Line 86: please, add references and the reason why Artemia was administered. Not all have to know how fish are feed!

Line 91: nothing is referred about the source of PSP. This means that it is not described anywhere from which portion of the plant (stem, roots, flowers etc...) the PSP was obtained nor how this part of the plant was treated to get the PSP.

Line 92: please, explain why Fluoxetine hydrochloride (FLU) has been used.

Paragraph 2.2.2 Line 100: The principle that methods must be described in order to be replicable by other researchers always holds true; so written, the paragraph is absolutely insufficient.

Line 157: which are the ‘appropriate’ modifications??

Line 166: FLU (FLU 0.1 mg/mL treatment 166 + CUMS). Which is the reason of this treatment? Non where it is explained..

Authors should somewhere also indicate why they have chosen the two 100 and 200mg/L concentrations.

Line 174: “..and fish in CON group were not subjected to any administration.” Do you mean any CUMS?

And:” ..Fish in CUMS group were immersed in 1system water without PSP or FLU?

Please, clarify.

Line 184: How many times it was performed the NNT ?

Line 243: are these primers for Zebrafish genes or mice genes?

Figure 3A Why FLU is not indicated?

The conclusion should be improved.

Who has validated the cd206, egr-2 and ym1 chitinase protein in zebrafish?  Please, add references.

Figure 6.  please, describe what is indicated by arrows in CUM picture?? And why nothing is described in Flu and PSP?

Line 442: In rodent CUMS procedure…

Line 447-448: Please, delete.

Lines 477-478: please, add references.

Line 480: the IOD significance must be explained with relative references.

Lines 562-564: please, add references.

Line 580-581: please, improve the sentence..

Minor comments

Line 67:Authors have added the definition of 5-HT and NE .

Line 440: please, correct ‘mimic’ with ‘induce’ and ‘a classic procedure’ (delete as a classic depression model)

Line 573: In the current study ??? Why, in others could change? Rephrase the sentence.

Line 574: Add Glc, Man, and Gal to the Abbreviation list.

Line 589: Data Availability Statement: The data that support the findings of this study are available from the 588 corresponding author upon request.

Comments on the Quality of English Language

Although I am not a native-speaking reviewer, I think there are some passages in the manuscript that should be reviewed by a native-speaking reviewer. The abstract, in particular, even compared to the rest of the manuscript, should be improved even in its 'construction'.

Author Response

Thanks very much for your hard work in reviewing our paper and the responses are attached. We hope it meet your requirements.

Reviewer 2 Report

Comments and Suggestions for Authors

Comments

ijms-2817255 Structural characterization and antidepressant-like effects of Polygonum sibiricum polysaccharides associated with regulating microglial polarization in CUMS-induced zebrafish

Overview: Authors have studied the anti-depressant effect of Polygonum sibiricum (PSP) in depressive-zebrafish model.

The manuscript is well-written but has some technical flaws which need to be addressed.

The manuscript cannot be published in the present form and needs Major revision.

Major comments

1.      Abstract: Line 15: Please give a brief info of what is CUMS.

2.      Material and methods: How PSP extract was prepared? The authors mentioned in line 91 that it was obtained from Shanghai Yuanye Bio-Technology. Does that mean the authors ordered dried extract from the company? Was it hydroalcoholic or aqueous extract and from which part of the plant?

3.      P 3, line 100: Please give the assay method in detail.

4.      P 6, L264-266: The statement is incorrect. The presence of phenol/flavonoid/proteins might exert synergistic/ additive/negative effects.

5.      Figure 1B. Label the two lines with different colors and mention which is which.

6.      Figure 1C: The axis scale is not clear.

7.      Table 2: Please mention the data units in the table. What does Mn,Mp,Mz stand for? Instead of a,b, c below the table please use the word abbreviation and explain Mn,Mw,Mp there. Also, please change the word “item” to another word.

8.      What is the EC50 of PSP in zebrafish?

9.      Does any ethical approval is required for the study? If yes, please mention the approval number at the end of the manuscript.

Minor comments

Line 15: reframe the sentence by omitting the word “firstly”

Line 153: change the word fish to zebrafish

Comments on the Quality of English Language

Minor editing of English language required

Author Response

We apprecite your carefulness in reviewing our paper and have revised them accoriding to your advice. The responses are attached. Thanks again.

Round 2

Reviewer 2 Report

Comments and Suggestions for Authors

The suggested corrections have been made in the manuscript.

However, I'm not satisfied with Table 1. On the right-hand column instead of the PSP, it would be better to mention "Units"

It can be accepted for publication after minor revisions in table 1.